# Coastal Nitrogen Drives Respiration Quotient in the Southern California Bight

Allison R. Moreno[1,2], Adam J. Fagan[3], and Adam C. Martiny[2,3]

**Affiliations:**
[1] Department of Ocean Sciences, University of California, Santa Cruz, CA, USA
[2] Department of Ecology and Evolutionary Biology, University of California, Irvine, CA, USA
[3] Department of Earth System Science, University of California, Irvine, CA, USA
*Correspondence to:* Allison R. Moreno (amoren53@ucsc.edu)

**Abstract.** Southern California Bight coastal waters are dynamic and strongly influenced by a changing climate. In the open ocean, an increased respiration quotient has been found during high temperature and low nitrogen conditions but variation in coastal environments is uncertain. To disentangle the controlling factors in a coastal environment, we examined environmental conditions, particulate organic matter, and the respiration quotient over five years in the Southern California Bight. Our study revealed clear seasonal variation in environmental conditions and biological parameters. We detected a higher than previously reported respiration quotient in open ocean regions. We found a strong inverse relationship between the respiration quotient, nitrate and chlorophyll. Our findings also suggest that changes in community structure, triggered by nutrient shifts and a local oil spill, affected the respiration quotient range and explains some of the variability measured. As climate continues to impact coastal regions, a variable respiration quotient is likely important for subsurface oxygen concentrations and in turn the health of our coastline.

## 1. Introduction

Oxygen is vital for coastal ecosystem health, acting as the primary oxidizing agent in cellular respiration. In the California Current, increased upwelling, mixing, and remineralization rates result in higher nutrient availability during winter and spring months (Venrick 2012), driving seasonality in primary production and subsequent dissolved oxygen demand in the subsurface layers (Bograd et al. 2008). Coastal deoxygenation events have been found to be primarily driven by nutrient cycling (Falkowski et al. 2011; Schmidtko et al. 2017). The respiration quotient or the amount of oxygen required for organic carbon oxidation, $r_{-O2:C,}$ is a potentially important factor for oxygen dynamics but is commonly assumed constant (see Methods for chemical equations). However, changes in the respiration quotient has been linked to surface plankton community composition and may be an additional regulator for deep ocean oxygen (Moreno et al. 2020). Increased $r_{-O2:C}$ is representative of larger oxygen consumption by bacteria or higher trophic organisms during respiration. The impact of an increased $r_{-O2:C}$ (1.3) away from Redfield (1.0) within an Earth System Model resulted in a 30% decrease in global oxygen rivaling impacts of a changing export production (Moreno et al. 2020; Gerace et al. 2023). Additionally, the amount of exported organic carbon and associated oxygen consumption due to bacterial respiration is sensitive to temperature and has a strong impact on oxygen levels (Matear and Hirst 2003; Keeling et al. 2010). As expected globally, temperatures will continue to rise in the Pacific Ocean, slowly warming the California coastline despite seasonal upwelling. However, the respiration quotient has not been quantified in coastal waters to date. Consequently, it is critical to quantify coastal $r_{-O2:C}$ dynamics and environmental controls.

Open ocean $r_{-O2:C}$ varies systematically with temperature and nutrients to produce distinct basin patterns. $r_{-O2:C}$ has been shown to vary positively with temperature in the Eastern Pacific Ocean (Moreno et al. 2020) and in nitrogen limited regions in the Atlantic Ocean (Moreno et al. 2022). Regions with high temperatures and deep nutriclines were found to have high $r_{-O2:C}$ ratios in the Pacific Ocean (Moreno et al. 2020). Further, nitrogen stress caused increased $r_{-O2:C}$ resulting in ratios higher than with temperature alone. In contrast, phosphorus stress appeared to dampen ratios, particularly when combined with high

temperatures (Moreno et al. 2022). Beyond the mixed layer, $r_{-O2:C}$ decreased to its minimum at the euphotic and disphotic boundary (Gerace et al. 2023). It was hypothesized that planktonic community structure and the preferential production/removal of biochemical components (i.e., lipid followed by proteins) led to a steady decrease in $r_{-O2:C}$ (Gerace et al. 2023). However, from the base of the euphotic zone to depth, the $r_{-O2:C}$ remained relatively consistent with the average higher than the mixed layer. Oxygen consumption was consistent, resulting in an overall decrease in the deep oxygen concentrations. Thus, $r_{-O2:C}$ vary systematically with environmental conditions in the open ocean and hence, we hypothesize that similar variation occurs in coastal regions.

The Southern California Bight (SCB) is ever dynamic and changing due to climatic influences creating a unique study site. Previous work, at the 'Microbes in the Coastal Region of Orange County' (MICRO) time-series, quantified variation in the particulate organic matter (POM) concentrations and stoichiometric ratios (C:N:P) corresponded to seasonal and multi-year oscillations in environmental conditions and phytoplankton abundances (Martiny et al. 2016; Fagan et al. 2019; Larkin et al. 2020) Specifically, high C:N:P corresponded to summer/fall periods with high temperatures, low nutrients and a small phytoplankton dominance and vice versa for cooler periods during the winter and spring (Martiny et al., 2025). Harmful algal bloom (HAB) forming species are present year-round, range in size and play a significant role in SCB biogeochemical cycling (Trainer et al. 2010). Changes in bloom behavior due to natural or anthropogenic influences will influence POM concentrations and in turn stoichiometric ratios. As such, we expect $r_{-O2:C}$ to exhibit seasonality and follow previously quantified stoichiometric ratio patterns with increased values during warm, low nutrient seasons, as well as major variation due to bloom formation and/or shifts in community structure.

In October 2021, an oil spill deposited over 20,000 gallons of crude oil onto Southern California beaches creating a natural community shift experiment (Brock et al. 2025). During this oil spill it was found that a significant beta-diversity shift occurred during this oil spill (Brock et al. 2025). Many other studies have found a high presence of hydrocarbon degrading bacteria during oil spills, shifting the microbial community structure away from seasonality (Hazen et al. 2010; Yang et al. 2014). Their magnitude is closely related to the season and type of oil during the event (Fuentes et al. 2016; Varjani and Gnansounou 2017). As a result, the bulk particulate matter (i.e. surface plankton community) will shift while oil persists in the water column and could cause a lag in seasonal community composition from pre-oil conditions. It has been hypothesized that changes to the bulk carbon type could alter the $r_{-O2:C}$ average ratio (Moreno et al. 2020). Provided with this unique opportunity affecting our study site, we expect to quantify decreased $r_{-O2:C}$ patterns while oil is present and possible shifts from seasonal or annual patterns.

The goal of this study was to quantify $r_{-O2:C}$ and identify possible drivers of its coastal temporal dynamics. To this end, we quantified changes in temperature, nutrients, POM concentrations and $r_{-O2:C}$ ratios at the MICRO time-series in the SCB weekly from October 2016 to January 2022. We predicted that the cumulative average $r_{-O2:C}$ will be higher than previously quantified open ocean ratios due to increased coastal dynamics and higher anthropogenic influence. We also predicted observing seasonal patterns in $r_{-O2:C}$, strongly controlled by temperature rather than changing in nutrient stress. As planktonic community composition shifts from larger to smaller species, we anticipated $r_{-O2:C}$ will increase. Finally, we expected that increased complex hydrocarbons during the oil spill period will decrease $r_{-O2:C}$ ratios.

## 2. Material and Methods
### 2.1. Seawater Collection
Surface water was collected weekly at the MICRO time-series study site (Fig. S1; 33.608˚N and 117.928˚W; Martiny et al. 2016) off the Newport Pier using a bucket. Two autoclaved bottles are rinsed with ocean water and filled for processing in the lab. Water temperature and chlorophyll a data are collected from an automated shore station off Newport Pier as part of the Southern California Coastal Ocean Observing Systems (SCCOOS).

For particulate organic carbon (POC) and for particulate chemical oxygen demand (PCOD) samples, we collect 300 ml for each analysis in triplicate from each bottle are filtered within an hour of collection through pre-combusted (500˚C, 5 hr.) 25 mm GF/F filters (Whatman, MA). Each filter is rinsed with Milli-Q water before sample filtration to remove potential P residues. Filters are stored in a starred petri dish. POC samples were placed directly into the -20˚C freezer. The PCOD samples are dried for 24 hr. at 55˚C and then placed in the -20˚C freezer. The filtrate from the initial filtration is collected and used for macronutrient quantification. The filtrate is filtered through a 0.2 µm syringe filter into a 50 ml tube. Triplicates were taken for nitrate and phosphate and stored in the -20˚C freezer.

## 2.2. Macronutrients Quantification

Soluble reactive phosphorus (SRP) concentrations were determined using the magnesium induced co-precipitation (MAGIC) protocol and calculated against a potassium monobasic phosphate standard (Karl and Tien 1992; Lomas et al. 2010). Nitrate samples (taken before 2019) were treated with a solution of ethylenediaminetetraccetate and passed through a column of copperized cadmium fillings (Knap et al. 1993). Nitrate samples (taken after 2019) were measured using a spongy cadmium method (Jones 1984).

## 2.3. Particulate Organic Carbon (POC)

After thawing, POC filters were allowed to dry overnight at 65˚C before being packed into a 25 mm tin capsule (CE Elantech, Lakewood, New Jersey). Samples were then analyzed for C content on the FlashEA 112 nitrogen and carbon analyzer (Therom Scientific, Waltham, Massachusetts), following the Sharp (1974) protocol. POC concentrations were calibrated using known quantities of atropine ($C_{17}H_{23}NO_3$) and acetanilide ($C_8H_9NO$).

## 2.4. Particulate Chemical Oxygen Demand (PCOD) Assay

PCOD samples were quantified following Moreno (2020). Samples were placed in 50˚C for 24 hr. Filters were then transferred to HACH HR+ COD vials (Product no. 2415915 containing mercuric sulfate). Two mL of Milli-Q water was added to each vial and inverted to submerge the filers completely. Vials are digested at 150˚C for 2 hr. in a digestion block. Samples were then cooled to room temperature. Due to uneven precipitation occurring, precipitation was induced by adding 92.1µL of 0.17M (or 9.5g L$^{-1}$) NaCl to each vial. Vials were inverted twice and centrifuged for 30 minutes at 2500 rpm and read on a photo-spectrometer at a wavelength of 600 nm. Note: Dichromate does not oxidize organic nitrogen, so this assay only quantifies changes in the carbon oxidation state. To quantify PCOD, in µM $O_2$, we utilize a standard curve based on HACH certified COD 1000 mg/L standard stock solution (Product no. 2253929).

## 2.5. $r_{-O2:C}$ Ratio

$r_{-O2:C}$ ratios were taken from the mean concentrations of PCOD and POC triplets. We compare our measured $r_{-O2:C}$ ratio to that of Redfield (1; Redfield 1958) and Anderson's best estimate average cell value of 1.1 (Anderson 1995). Using Equation 1 and 2, where the nitrogen end member is ammonia, the $r_{-O2:C}$ for Redfield (106:263:110:16:1; C:H:O:N:P) is 1 and 1.1 for Anderson (106:175:42:16:1; C:H:O:N:P).

$$C_x(H_2O)_w(NH_3)_yH_zH_3PO_4 + \left(x + \frac{1}{4}z\right)O_2 \rightarrow xCO_2 + yNH_3 + H_3PO_4 + \left(w + \frac{1}{2}z\right)H_2O \quad (1)$$

$$r_{-O2:C} = \frac{\left(x + \frac{1}{4}z\right)}{x}. \quad (2)$$

The standard deviation for $r_{-O2:C}$ were calculated as a pooled sample:

$$\sigma_{r-O2:C} = \frac{-O_{2,aver}}{C_{aver}} \times \sqrt{\left(\left(\frac{\sigma_{-O2}}{-O_{2,aver}}\right)^2 + \left(\frac{\sigma_C}{C_{aver}}\right)^2\right)}$$

**2.6. Large Phytoplankton Relative Abundance**
Phytoplankton abundance for ten species, two general species categories and the total
phytoplankton abundance were obtained from SCCOOS. Water collection and analysis can be found at
(Seubert et al. 2013). The ten species are as follows: *Akashiwo sanguinea, Alexandrium spp., Dinophysis*
*spp., Lingulodinium polyedra, Prorocentrum spp., Pseudo-nitzschia (PN) delicatissima, PN seriata,*
*Ceratium spp., Cochlodinium spp.,* and *Gymnodinium spp.* The two categories are 'Other Diatoms' and
'Other Dinoflagellates' (Table 1). Monthly and annual average relative abundance were calculated by
averaging the abundance for each species over the specific time span (either monthly or annually) then
dividing by the total abundance for that same period.
**2.7. Statistical Analysis**
All analyses were done using Table S1 data in Matlab (Mathworks, MA). Using the smooth
function, a four-point or eight-point moving average was overlaid onto the raw data time-series plots.
Sum of square analysis was conducted on linear regressions to quantify the monthly and annual
contributions. To detrend seasonality in our time-series parameters, we apply a season adjustment using a
stable seasonal filter applying a 53-point moving average, representing our weekly sampling. To quantify
statistical significance in monthly and annual differences, a 1-way ANOVA is used. To determine
potential covariations, a Pearson's correlation coefficient was calculated for each pair of environmental
variables, followed by a test of statistical significance (p-value $\leq 0.05$). Similarly, a Pearson's correlation
coefficient was calculated between each species abundance and chlorophyll, followed by a statistical
significance (p-value $\leq 0.05$) test, to determine which species influenced chlorophyll concentrations.
To determine impacts on $r_{-O2:C,}$ a Pearson's correlation coefficient was calculated for each pair of
species relative abundance and $r_{-O2:C,}$ followed by a statistical significance test (p-value $\leq 0.05$). Due to
limited data in the large phytoplankton relative abundance, we removed *Ceratium spp., Cochlodinium*
*spp.,* and *Gymnodinium spp.* from the annual correlation analysis.
Statistical nonlinear models were fitted using six predictor variables (temperature (˚C), nitrate
(μM), phosphate (μM), chlorophyll (mg C/m$^3$), POC (μM), and PCOD (μM)). $R^2$ and Akaike Information
Criterion (AIC) were used to compare across models. For all regressions containing interpolated
parameters, a random sampling of cruise data was conducted to ensure results were not swayed.
Oil spill analysis was accomplished using a 2-way ANOVA. First $r_{-O2:C}$ ratios were averaged over
three months before the spill (July through September), the month during the spill (October), and three
months after (November through January) during 2021/2022. Similar averaging was done for 2018/2019
and 2019/2020 years to quantify differences between the average $r_{-O2:C}$ ratios during this period and
specifically during the oil spill.
**3. Results**
To evaluate our hypothesis that $r_{-O2:C}$ demonstrates seasonal variability, and has systematic
relationships with environmental conditions, we quantified physical (temperature), chemical (nitrate and
phosphate), and biological [chlorophyll, particulate organic carbon (POC), particulate chemical oxygen
demand (PCOD), and plankton relative abundance] properties and stoichiometric ratio ($r_{-O2:C}$) over a 5-yr
period from 2016 to 2022. Annual oscillations and strong correlations exist between parameters.
**3.1. Temporal Patterns**
Physical conditions demonstrate short-term, seasonal, and annual trends. As expected for a site at
33˚N, temperature oscillated annually with a peak in August and trough in January (Fig. 1A). The highest
average temperature occurred in 2018 with an annual mean of 18.05˚C. As described previously, nutrient
availability showed a strong seasonal anti-correlation with temperature (Martiny et al. 2016; Fagan et al.
2019) as well as clear monthly and annual differences (ANOVA $p < 0.05$; Fig. S2). Temperature
correlations with other parameters differ on weekly, monthly, and annual timescales (Fig. S3).
Temperature dynamics covaried with macronutrients. Macronutrient concentrations demonstrate clear
patterns. Phosphate concentrations appeared to have a 3-year systematic shift, going from approximately
0.2 µM and steadily increasing to approximately 0.6 µM (Fig. 1B). This pattern was seen from 2016 to
2019, and again from 2019 to 2022, with a quick drawdown occurring winter 2018/2019 corresponding to
an increase in chlorophyll. Phosphate was highest during the winter months, and lowest during the late
summer. Weekly phosphate was correlated with temperature and nitrate (Fig. S3A). Nitrate
concentrations shift on annual and seasonal cycles (Fig. 1C). Generally, nitrate was highest (with the least
amount of variation) in 2018 and lowest (with the highest variation) in 2022 (Fig. S2). Nitrate was
correlated with weekly temperature, phosphate, and r$_{-O2:C}$ (Fig. S3A). Nitrate and phosphate, on monthly
scales, were correlated with each other, and temperature (Fig. S3B). In late 2017 to 2019, nitrate and
phosphate were consistently higher than during the late 2019 to 2021 timeframe. Chlorophyll follows a
seasonal cycle with peaks during period of low nutrients, possibly responding their drawdown of nutrients
for growth. Dynamic environmental conditions at MICRO could have strong impacts on biological
parameters leading to distinct patterns.

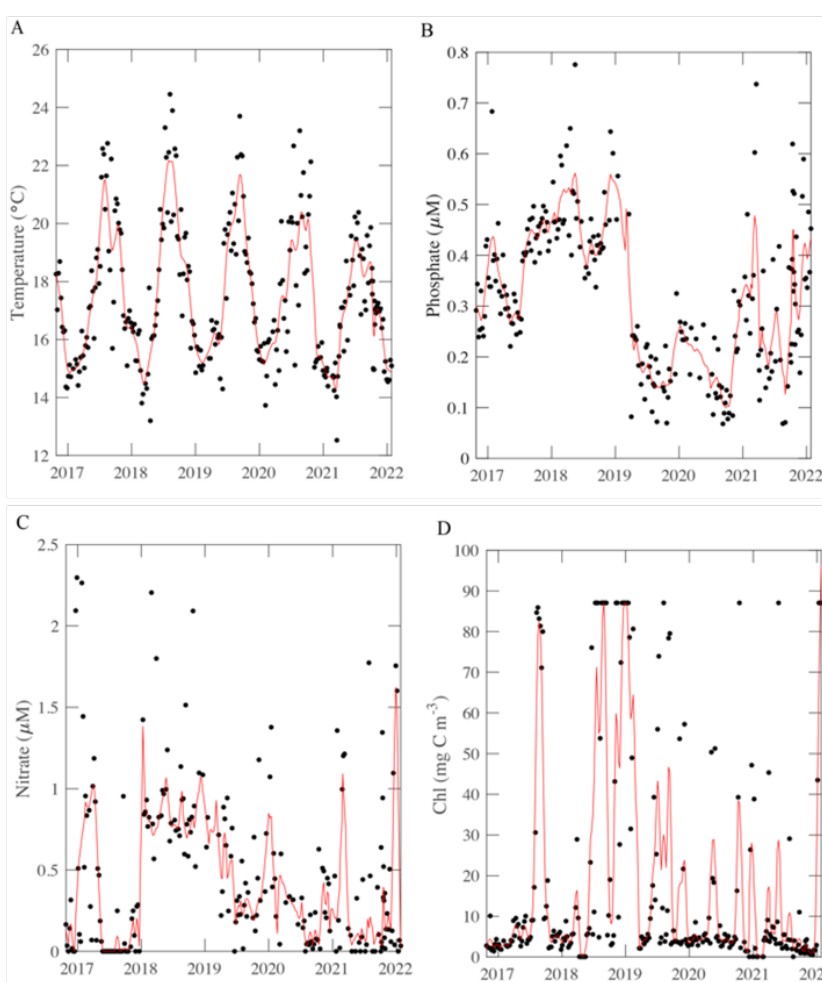

**Figure 1. Environmental conditions, macronutrient, and chlorophyll concentrations (A- D) over time at**
**MICRO study site in Newport Pier, Newport, CA.** The solid black points represent the average data per week
from the period 10/26/2016 to 1/31/2022. The red line represents an 8-point moving average.

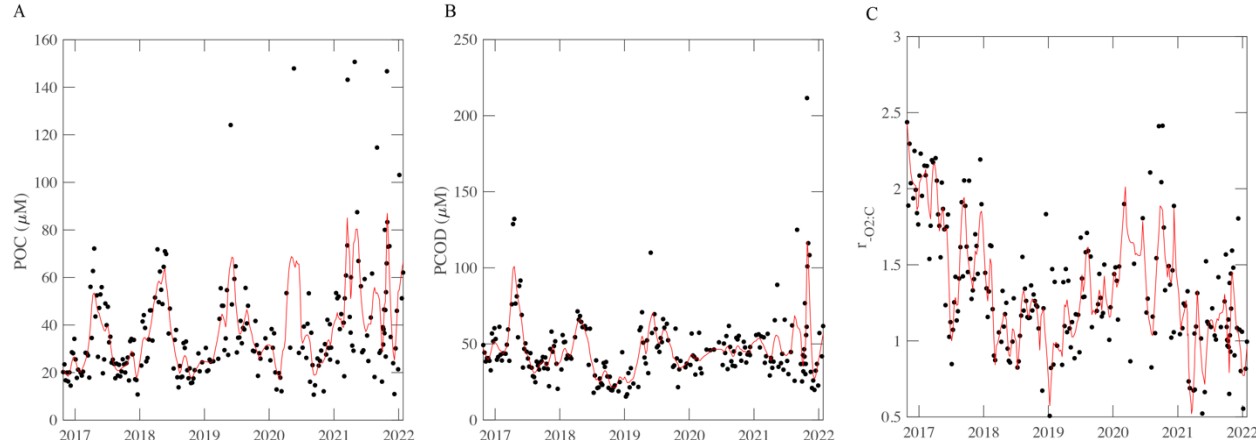

**Figure 2. POM concentrations (A, B) and respiration quotient (C) over time at MICRO study site.** The solid black points represent the average data per week from the period 10/26/2016 to 1/31/2022. The red line represents an 8-point moving average for A and B, and a MATLAB 'rlowess' for C moving average. The respiration quotient is a molar ratio.

POM concentrations also demonstrated short-term, seasonal and annual variation. [POC] and [PCOD] concentrations peaked during the spring bloom period (May) and oscillated annually (Fig. 2A, B and Fig. S2, S3). Generally, [POC] had similar variability to [PCOD] ($r^2 = 0.70$ and p-value < 0.05; Fig. 2A, B). [POC] appeared to be increasing through time with the highest annual average occurring in 2022. [POC] correlated with physical conditions (Fig. S3). [PCOD] was significantly higher in April and May (Fig. S2). [PCOD] also had its highest annual average in 2022. Annually [PCOD] covaried with temperature (Fig. S3C). However, where [POC] was lowest in 2020, [PCOD] had a minimum in 2018 (Fig. S3). Overall, biological parameters showed similar multiannual oscillations as environmental conditions, indicating consistency between the two.

### 3.2. Coastal $r_{-O2:C}$ ratios

The respiration quotient showed clear temporal variation but with no significant seasonality. The average $r_{-O2:C}$ ratio was 1.34, which was statistically higher (t-test, p-value < 0.05) than Redfield (1), Anderson (1.1), and previous open ocean estimates (1.16; Moreno et al. 2022). Although the average was higher, the range in values was smaller compared to open ocean samples. Generally, the highest monthly $r_{-O2:C}$ were found in February and September (Fig. S4). However, the highest weekly $r_{-O2:C}$ peak occurred in January and into February 2020, when [POC] values were low (Fig. 2C). The lowest average annual $r_{-O2:C}$ (0.99) was measured in 2022. Additionally, a multi-year (~3 year) trend was detected, whereby $r_{-O2:C}$ decreased from 2016 to 2020 and concentrations peak at 1.85 (Fig. 2C). Weekly nitrate and [POC] were found to correlated with changing $r_{-O2:C}$ (Fig. S4). We found distinct variation in the respiration quotient ratio at the MICRO site.

We assessed links between $r_{-O2:C}$ environmental changes using a multi-dimensional principal component analysis (Fig. 3). As shown above, changes in environmental conditions covary and be associated with changes in particulate organic matter (Fig. S3). We explicitly accounted for this co-variance using principal components (PC) and linked each PC to $r_{-O2:C}$ variation. Our PC analysis explained ~81% of the $r_{-O2:C}$ variance. The environmental parameters used within the multi-dimensional analysis were temperature, phosphate, nitrate, and chlorophyll concentrations. Our first principal component (PC1) represented blooming conditions (i.e., positive chlorophyll) representing 50% of overall variance. PC2 captured the environmental axis, dominated by seasonality (~31%; Fig. 3). Thus, PC2 corresponded to a decreased temperature with negative values and increased nutrient concentration with positive. Specifically, high $r_{-O2:C}$ ratios can be found under low nitrate and chlorophyll concentrations. Similar to open ocean analysis, when only considering a single environmental parameter, nitrate concentrations control $r_{-O2:C}$ such that increased $r_{-O2:C}$ is present during nitrogen stress (p-value = 0.0207,

$r^2 = 0.023$). Our analysis demonstrated an anti-correlation with environmental conditions which suggests
that higher $r_{-O2:C}$ is present under high temperature and low nutrient conditions. This corresponds to
blooming behavior. As nutrients stimulate bloom formation $r_{-O2:C}$ was low, and as blooms dissipate, the $r_{-O2:C}$
$_{O2:C}$ increased. Hence, $r_{-O2:C}$ is shifting based on the environmental changes.

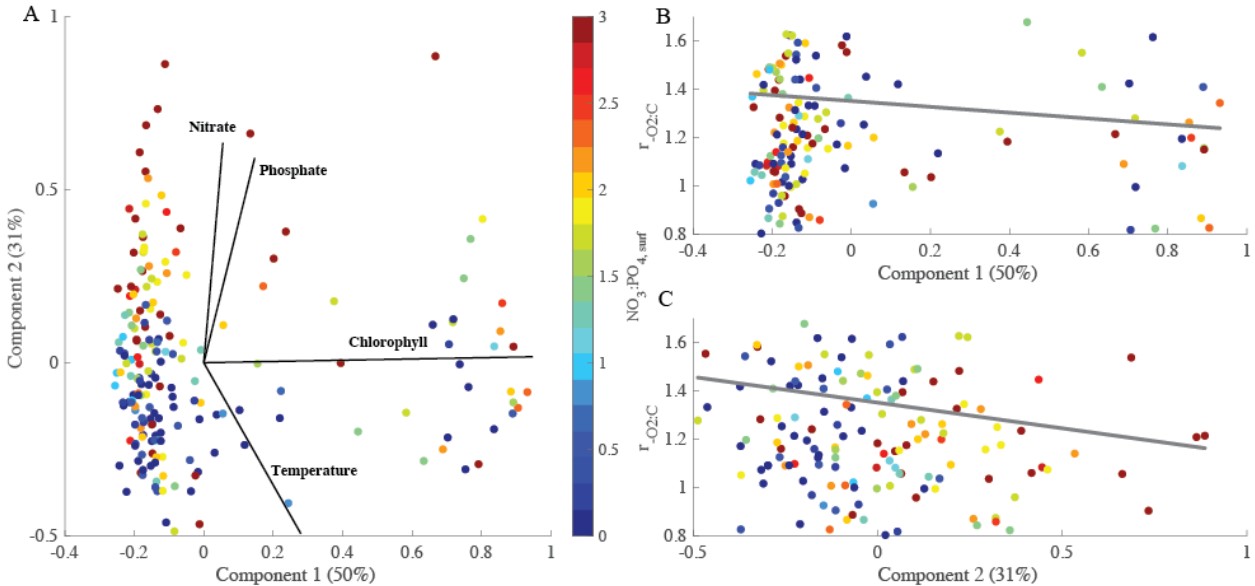

**Figure 3. PCA analysis of $r_{-O2:C}$ to determine the overall controlling factors**. A. PCA of four environmental
variables (nitrate concentrations, phosphate concentrations, chlorophyll concentrations, and temperature) over 5-
years at the MICRO site. B. and C. $r_{-O2:C}$ explained by the first and second principal component including the $r_{-O2:C}$
based on linear regression analysis. The percentages of total variance represented by principal component (PC) 1
and 2 are shown in parentheses. Colored dots represent the $NO_3:PO_{4, surf}$ concentration. Grey solid lines represent
regression lines with $r^2 = 0.0003$ (B) and 0.012 (C).

**3.3. Large Phytoplankton Abundance**
Large harmful algal bloom (HAB) phytoplankton abundance demonstrated clear shifts over the
time-series. California experienced seasonal and environmentally driven HAB blooms. Ten HAB species
and two categories of large phytoplankton (Table 1) were quantified and compared over our time-series to
determine if community structure plays a role in $r_{-O2:C}$ variation. Seasonal cycling can be observed in
*Prorocentrum*, *Pseudo-nitzschia (PN) delicatissima*, *PN seriata*, 'Other Diatoms', and 'Other
Dinoflagellates' (Fig. S5). Whereas other species, i.e. *Akashiwo*, *Alexandrium*, *Cochlodinium* and
*Gymnodinium*, are observed sparingly. On weekly timescales, *Akashiwo* had a positive relationship with
chlorophyll a concentration (p-value = 0.004, r = 0.03), whereas 'Other Dinoflagellates" has a negative
relationship (p-value = 0.032, r = -0.02). On monthly timescales, the 'Other Diatoms' and 'Other
Dinoflagellates' made up 50 to 80% of total phytoplankton (Fig. 4A). Following this majority, we
observed significant concentrations of *Prorocentrum*, *PN delicatissima* and *PN seriata*. During typical
blooms months (spring and August) we estimated a higher presence of species, with an increase in a few
species during these months. Similarly, annual relative phytoplankton abundance observations were
highly dominated by 'Other Diatoms' and 'Other Dinoflagellates' (Fig. 4B). *Akashiwo*, *Alexandrium*, and
*Dinophysis* were found in very low concentration throughout the time-series, however, evidence shows
their impact could be great even so. Species sampling was paused from March to June 2020, which may
explain the lowest diversity compared with other years. In 2021, the 'Other' categories were at their
lowest at 40% of the total phytoplankton present. Additionally, in 2021 and 2022, *Lingulodinium*'s
relative abundance becomes a significant part of the breakdown at approximately 15 to 20%.
Accordingly, we observed clear shifts in the phytoplankton community composition and corresponding
relative abundance of larger cell volume species over our time-series.
**Table 1. Quantified genius and category of large harmful algal bloom phytoplankton.**

| Type | Genius/Species | Size Range |
|---|---|---|
| Dinoflagellates | *Akashiwo sanguinea* | 40 – 80 μm |
| Dinoflagellates | *Alexandrium spp.* | 20 – 80 μm |
| Dinoflagellates | *Dinophysis spp.* | 35 – 50 μm |
| Dinoflagellates | *Lingulodinium polyedra* | 40 – 54 μm |
| Dinoflagellates | *Prorocentrum spp.* | 40 – 50 μm |
| Diatoms | *Pseudo-nitzschia delicatissima* | 40 – 80 μm |
| Diatoms | *Pseudo-nitzschia seriata* | 105 - 115 μm |
| Dinoflagellates | *Ceratium spp.* | 20 – 200 μm |
| Dinoflagellates | *Cochlorinium spp.* | 30 – 50 μm |
| Dinoflagellates | *Gymnodinium spp.* | 40 – 75 μm |
| Diatoms | 'Other Diatoms' | 20 – 200 μm |
| Dinoflagellates | 'Other Dinoflagellates' | 5 – 2,000 μm |

Shifts in community structure impact $r_{-O2:C}$ dynamics. *Akashiwo sanguinea*, on a weekly basis,
was found to positively correlate with $r_{-O2:C}$ (r = 0.18, p-value <0.05) and 'Other Dinoflagellates' were
found to have a negative relationship (r = - 0.14, p-value <0.05). Mean $r_{-O2:C}$ is associated with months
that contain more diverse relative abundance in large phytoplankton (January, April, and August; Fig. S4
and Fig. 4). In March and May, 'Other Diatoms' and 'Other Dinoflagellates' decline in abundance, and
this correlated to months with the lowest $r_{-O2:C}$. Comparatively, during these months *Pseudo-nitzschia*
species are in highest relative abundance. This result is expected as upwelling replaces the surface with
cooler nutrient rich waters stimulating plankton growth. In January, *Akashiwo sanguinea* is present in
observable concentrations and associated with lower $r_{-O2:C}$ (Fig.4 and Fig. S6A). *Akashiwo sanguinea* are
similar in size to *Pseudo-nitzschia delicatissima*, which may provide evidence of a relationship between
cell size and variation in $r_{-O2:C.}$. We found additional general trends which provide a possible line of
evidence for future work (Fig. S6B). In 2016 and 2020, the relative abundance of 'Other Diatoms' and
'Other Dinoflagellates' is approximately 80% of the total and was associated with high $r_{-O2:C}$ values (Fig.
4 and Fig. S4). However, in years with high distinct diversity (2019 and 2021), where 'Other Diatoms'
and 'Other Dinoflagellates' relative abundance is low, the average $r_{-O2:C}$ is low (Fig. 4). As such, we
demonstrate that community structure shifts can play a role in the average $r_{-O2:C}$ and its variability.

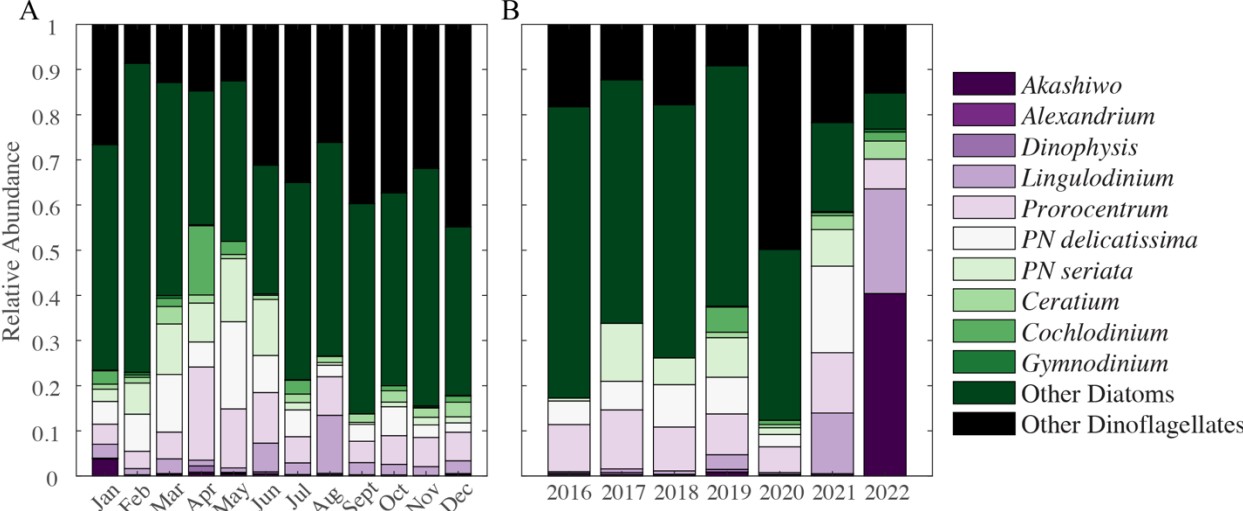

**Figure 4. Monthly (A) and annual (B) large phytoplankton relative abundance.** Each color represents a
different species, genus, or category of diatom or dinoflagellate.
**3.4. Orange County 2021 Oil Spill**
The oil spill event had an impact on $r_{-O2:C}$. There was no immediate response in $r_{-O2:C}$ to the oil
spill following the first day of its occurrence (Fig. 5). However, in the following days (approximately a 2-
day later), an increase in the [POC] and [PCOD] persisted until the beginning of November. However,
there is a slight discrepancy between [POC] and [PCOD] in the first few November measurements (Fig.
5). [POC] continued to peak in the first week before starting to come down, whereas [PCOD] started to
decrease immediately. $r_{-O2:C}$ show similar trends- increases after a 2-day lag. $r_{-O2:C}$ recovery is
complicated to quantify; however, it appears that values are slightly lower but within range of pre-oil spill
concentrations. Statistically, there is no difference in average $r_{-O2:C}$ between 3-months before, 1-month
while oil was present in samples (during), and 3-month after. Intriguingly, when compared with other
years (2019, 2020, and 2021), the 2021 oil spill year did have overall lower $r_{-O2:C}$ averages (Fig. S7). $r_{-O2:C}$
in 2020 has the highest concentrations for this comparative period. However, due to the range within the
time periods, there is no statistical annual differences. Thus, the oil spill does appear to influence $r_{-O2:C}$
variability, though more research is needed to fully disentangle the mechanistic controls.

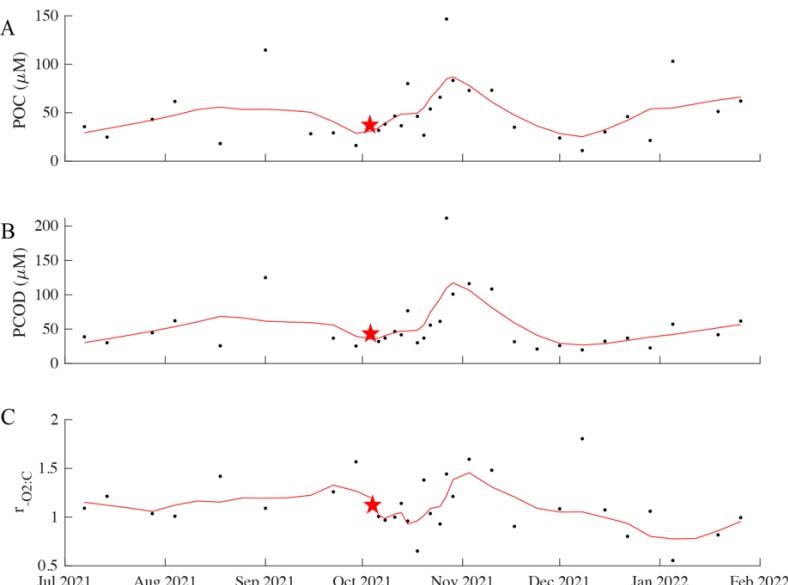

**Figure 5. Biological parameters (A. POC, B. PCOD, and C. $r_{-O2:C}$) from July 1, 2021, to January 31, 2022.**
This represents a 3-months before to 3-months after the oil spill occurrence. The red line represents an 8-point
moving average. The red star marks the start of the oil spill.

**4. Discussion**
We find evidence that nutrient limitation controls the $r_{-O2:C}$ patterns in the coastal MICRO study
site. Direct quantifications of environmental parameters, large HAB forming phytoplankton relative
abundance, and the $r_{-O2:C}$ allowed us to disentangle patterns and hypotheses to determine the relative
control of abiotic vs. biotic processes in a coastal environment. It was previously proposed that
temperature and nutrient availability are the two main environmental controls that have been shown to
influence the respiration quotient in surface open ocean communities (Moreno et al. 2020, 2022). Within
this dynamic location, we have quantified a larger average ratio ($1.34_{0.51}^{2.43}$) compared to open ocean
samples ($1.16_{0.75}^{1.93}$; Moreno et al. 2022). This is relatively equivalent to an increase in 18% of oxygen
consumed per organic carbon in coastal waters. We find that low nitrate and chlorophyll concentrations
correspond to a higher [PCOD] and lower [POC] resulting in higher $r_{-O2:C}$ (Fig. 1 and 2). Similar to
previous open ocean shifts, nitrogen stress corresponded to higher ratios (Moreno et al. 2022). A higher $r_{-}$
$_{O2:C}$ could lead to a decreased oxygen content deeper in the water column, as surface particles sink. Many
microalgae under N stress will increase their lipid content (Thompson et al. 1992; Reitan et al. 1994;
Juneja et al. 2013) consequently increasing their average $r_{-O2:C}$. This is one possible line of evidence as to
why we found an increased ratio. Changes in nutrients will result in a physiological response that can
change the chlorophyll concentrations for the overall community; it can also shift the community
structure toward species that require less N. Our results indicate an interaction between N stress and
chlorophyll a, which corresponds to a response in $r_{-O2:C}$.
Our data suggests that $r_{-O2:C}$ is also directly influenced by the community structure (Fig. 4). Large
phytoplankton have been hypothesized to have a lower $r_{-O2:C}$ on average compared to small phytoplankton
(Moreno et al. 2022). We estimated the average functional group $r_{-O2:C}$ based on an allometric relationship
with molecular components- primarily lipid, carbohydrate, and protein concentrations (Moreno et al.
2022). Here, we found that $r_{-O2:C}$ was high in years where relative abundance of 'Other Diatoms' and
'Other Dinoflagellates' are high (2016 and 2020). An increase in lipid storage will increase their $r_{-O2:C}$
range and add variability. Although dinoflagellates are estimated to have an average $r_{-O2:C}$ (average size
and protein rich flagella), they also have storage capabilities. Similar to diatoms, during stress an increase
in lipid storage in dinoflagellates will increase their $r_{-O2:C}$ range. Shifts in physiological response by larger
phytoplankton under stress could impact variable $r_{-O2:C}$ in this region. Additionally, changes in
cyanobacterial ecotypes could also affect $r_{-O2:C}$. Previous time-series studies of cyanobacteria have
demonstrated interannual patterns in ecotype relative abundance and seasonal switching in ecotype
occurring in response to rapid environmental changes (Tai and Palenik 2009; Malmstrom et al. 2010;
Nagarkar et al. 2018; Larkin et al. 2020). Although ecotypes are genetically similar, physiological and
morphologically differences exist. Additionally, intracellular metabolites that are involved in methylating
DNA, RNA, and proteins differ in culture and effect metabolism differently per ecotype (Kujawinski et
al. 2023). Under nutrient replete conditions *Prochlorococcus* and *Synechococcus* C:P and N:P ratios
range from 121-165 and 21-33 (Bertilsson et al. 2003), demonstrating significant variation in cellular
composition. Further,  cellular quotas of *Prochlorococcus* and *Synechococcus* in the North Atlantic varied
significantly demonstrating species diversity and size distributions within the surface community (Baer et
al. 2017). Although we do not quantify cyanobacteria in this study, previous published work at the
MICRO site with a short overlapping window agrees with an ecotype shift being captured (Larkin et al.
2020). Our 2016 to 2018, $r_{-O2:C}$ ratios overlap with the Larkin data (Fig. 4) and as expected $r_{-O2:C}$ is
highest in 2016 when *Prochlorococcus* HLI is in highest abundance. As *Synechococcus* IV dominates the
region, the $r_{-O2:C}$ ratio decreases. *Prochlorococcus* HLI dominate the cyanobacterial community under
high temperature and lower nitrate concentration, whereas *Synechococcus* IV dominate in lower
temperatures and higher nitrate. As *Synechococcus* IV dominates the region, the $r_{-O2:C}$ ratio decreases. The
shift in dominating cyanobacteria under it optimal conditions could result in a shift in the bulk chemical
composition, in turn the POC concentration and $r_{-O2:C}$. We suggest that a shift in environmental conditions
will impact the surface community composition and could add variation to $r_{-O2:C}$ .
Trophic structure and detrital matter could also influence $r_{-O2:C}$ variability. Zooplankton and
heterotrophic bacteria can makeup ~50% of surface organic matter (Martiny et al. 2016). The relative
presence and abundance of heterotrophic organisms are important in breaking down particles and setting
POC concentrations. For example, POC is comprised of ~5 to 30% lipids (Wakeham et al. 1997; Hedges
et al. 2001; Behrendt et al. 2024) and up to 25% detrital matter (Kaiser and Benner 2008; Martiny et al.
2016). In a recent study, bacterial degradation rates differed based on lipid content and bacterial
preference (Behrendt et al. 2024). Some bacteria were highly selective and targeted specific lipid classes,
whereas others did not (Behrendt et al. 2024). Preferential remineralization can occur (e.g., particulate
organic phosphorus (Gundersen et al. 2002) and phosphonates (Benitez-Nelson et al. 2004). While the
physiological mechanisms for this preferential remineralization are still being researched, its presence
could have strong impacts on the fate of detrital matter in surface waters, particularly in dynamic
coastline. These variations in POM remineralization and thus, oxygen consumption, can be reflected in
the $r_{-O2:C}$. Similarly, the quality and quantity of detrital matter captured in the POM will also have an
impact. Although, regional shifts in the fraction and quality of the detrital matter are currently unknown,
previous estimates found that detritus and terrestrially derived particles make-up a small contribution of
POC (Martiny et al. 2016). However, as grazers messily breakdown organic matter, produce fecal pellets,
and reproduce, the composition of POC is changing. A shift in organic C caused by trophic interactions
and/or detrital matter will result in a one directional shift in $r_{-O2:C}$.
Crude oil does not directly affect $r_{-O2:C}$ ratios but appears to indirectly affect the community
dominance. $r_{-O2:C}$ do not show a significant impact compared with a three-month period before and after
(Fig. 5 and Fig. S6). Instead, a two-day lag in $r_{-O2:C}$ ratios exist (Fig. 5), which could be attributed to a
community response to the amount and type of oil present. Hydrocarbon degrading bacteria are always
present in small concentrations in most coastal regions, especially in highly anthropogenically impacted
coastlines. Incubation experiments found large significant shifts in both bacterial and archaeal
communities in seawater (Aktas et al. 2013). After the initial oil spill day, our [PCOD] and [POC]
increased resulting in a slight $r_{-O2:C}$ increase. This response could be due to the continued increase in oil
concentrations and its effect on the community. Increased oil can preferentially cause certain bacterial and
archaeal species to bloom and affect the rate of degradation- some compounds will degrade quickly and
others extremely slowly (Leahy and Colwell 1990). As the oil is degraded to low concentrations (or
background values in our coastal setting), $r_{-O2:C}$ ratios taper off to normal values and behaviors. Although
$r_{-O2:C}$ does not show large shifts during the oil spill, there is evidence that oil presence does indirectly
impact $r_{-O2:C}$ ratios through community shifts and their physiological responses.
There are multiple caveats to be considered within this study that could affect the overall
findings. The most prominent and obvious is the extremely high respiration quotient ratios. We quantify
ratios that are higher than previously measured and expected. Within our data, higher ratios typically
corresponded to higher [PCOD] rather than a lower [POC]. One possibility that needs to be explored in
the future is if our [PCOD] assay is capturing the oxidation of iron (Fe) and/or biogenic sulfur (S) on
samples. Trace concentrations of either would be difficult to explicitly quantify from samples taken.
Dissolved Fe is a main limiting nutrient along the California Current system (Hutchins et al. 1998).
However, iron from continent margin sediments have been shown to affect primary production and
carbon export in the Pacific Ocean (Johnson et al. 1999; Lam et al. 2006). A small fraction of the
sediment-derived Fe remains in solution as organic ligands (Kondo and Moffett 2015; Homoky et al.
2021) or in suspension as colloids or nanoparticles (Pan et al. 2011; Krachler et al. 2012). Sulfur, on the
other hand, makes up about 1% of organismal dry weight and is rarely limited in the ocean. There is a
large variation in its oxidation state, ranging from completely reduced (-2) to completely oxidized (+6).
Inorganic sulfur compounds thiosulfate and sulfite can be transformed to hydrogen sulfide and sulfate by
bacteria making these S species readily available. Additionally, organic hydrogen sulfides like sulfur-
containing amino acids, dimethylsufoniopropinate and 2,3-dihydroxypropane-1 sulfonate are also highly
present in surface waters and play vital roles in sulfur cycling (Hu et al. 2018). In natural respiration
processes, Fe reduction precedes sulfate reduction. However, within our assay dichromate is a powerful
oxidate and could be used to oxide reduced Fe and S, resulting in high $r_{-O2:C}$. Although this is still to be
considered, [PCOD] is more constrained than [POC] throughout the time-series (Fig. 2), so if we are
oxidizing either element, it is being done uniformly. Another caveat we recognize is the lack of sample
blanking, which could introduce some bias. However, previously we touched upon the idea that sample
blanking is not important for open ocean samples because of the high volume filtered (Moreno et al.
2022). Due to high biomass within coastal waters, the filtered volume within this study is smaller (300
mL compared to 2 L vs 8 L). The minimal variation in [PCOD], provides evidence that blanking would
also not play a strong role in quantifying the $r_{-O2:C}$. Our third caveat is the presence of non-biological
material (i.e., sand or dirt). During periods of higher biomass, darker filters are observed. High biomass
filters can become packed or overloaded, allowing for higher rates of excess biological and non-biological
material (including nanoparticles) present on filters. Note that darker filters are not equivalent to higher
ratios. Although there are a few recognized caveats, our findings are bringing new insights into coastal
variation and are robust.
**5. Conclusions**

Our study suggests that the $r_{-O2:C}$ is controlled by plankton's response to nutrient stressors in this coastal ecosystem. High $r_{-O2:C}$ along the coast has very strong implications on future hypoxic region expansions and general ecosystem health. Here, increased $r_{-O2:C}$ could lead to increase oxygen consumption by grazers and bacteria. Given future rising surface ocean temperatures (Di Lorenzo et al. 2005; Durack et al. 2018; Kwiatkowski et al. 2020; Xu et al. 2022) and continued changes to nutrients, $r_{-O2:C}$ dynamics further the complexities of coastal deoxygenation. Both direct (nutrient concentrations) and indirect (community shifts) controls play important roles in setting $r_{-O2:C}$ and therefore future oxygen levels. Although much research is still needed to understand the mechanistic responses to nutrients and its effects on coastal oxygen levels, changes to community structure appear to have an impact on variable $r_{-O2:C}$ in this dynamic region. As shown in our study the average $r_{-O2:C}$ is 18% higher than open ocean measurements. With increased temperatures and decreased nutrients due to climate change, we may observe increased coastal deoxygenation events at a higher frequency than in the open ocean. The potentially devastating impacts of increased events threaten the health and livelihood of surrounding communities that rely to tourism, fishing, and other ecosystem services. The respiration quotient is a grazer proxy for oxygen consumption during the breakdown of organic C. Ultimately, the MICRO $r_{-O2:C}$ is higher than previously measured open ocean samples. Our findings have strong implications for future respiration and oxygen cycling. Future research is needed to examine other coastal environments under a changing climate to understand the severity of coastal deoxygenation and determine if this study site is anomalous.

**Author contribution:** ARM and ACM designed the time-series. ARM and AJF collected and analyzed samples. ARM prepared the manuscript with contributions and edits from all co-authors.

**Competing interests**: The authors declare that they have no conflict of interest.

**Acknowledgements:** We thank all undergraduate students through the years that participated at the MICRO time-series, as well as Alyse Larkin for their assistance in editing our manuscript.

**Financial support:** NSF GRFP, UCI Chancellor's Club Fellowship, and the UCLA Chancellor's Postdoctoral Fellowship to ARM as well as NSF OCE-2135035 and NSF-OCE 1948842 to ACM.

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
