# Peer review of "Coastal Nitrogen Drives Respiration Quotient in the Southern California Bight"

_EGUsphere, 2025_

## Author Response (AR1)

AUTHOR REPONSE

>>> We have attempted to address all comments directly and hope the editor and reviewers find this revised version improved.

**Reviewer # 1:**

Dear Authors,

This study collected long-term samples of suspended particulate organic matter, phytoplankton community composition, and nutrients. Temporal variations were observed across different years. Due to an accident, the authors compared the relationships among these parameters and ultimately assessed the influence of temperature on phytoplankton growth. This type of multi-parameter, long-term dataset is valuable and should be supported, as it provides new insights into biogeochemical processes and ecosystem dynamics. However, certain aspects—particularly the interpretation of elemental ratios— warrant further clarification and deeper discussion. My major comments are as follows.

AUTHOR REPSONSE

>>> We appreciate the concerns and comments to improve our manuscript.

Major Comments

1. Redfield Ratio Clarification (Line 132)

Why is the Redfield ratio cited as "1" in line 132? Please explicitly state the formula being used (e.g., -$O_2$:C or N:P), and clarify the basis for this value.

AUTHOR REPSONSE

>>> In our chemical assay, ammonia is not further oxidized to nitrate. As such, our calculation for $r_{-O2:C}$ assume nitrogen is released as ammonium (see below). When using the C:H:O:N:P ratios for Redfield and Anderson proportions in Equation 1 and 2, $r_{-O2:C}$ is 1 and 1.1, respectively. We added the following text to the manuscript to make this clear: "Using Equation 1 and 2, where the nitrogen end member is ammonia, $r_{-O2:C}$ for Redfield (106:263:110:16:1; C:H:O:N:P) is 1 and 1.1 for Anderson (106:175:42:16:1; C:H:O:N:P).

$$C_x(H_2O)_w(NH_3)_yH_zH_3PO_4 + \left(x + \frac{1}{4}z\right)O_2 \rightarrow xCO_2 + yNH_3 + H_3PO_4 + \left(w + \frac{1}{2}z\right)H_2O \ (1)$$

$$r_{-O2:C} = \frac{\left(x + \frac{1}{4}z\right)}{x}, \hspace{4cm} (2)"$$

The study reports elemental ratios exceeding the canonical Redfield values, but the associated biogeochemical implications are not clearly explained. Although the ratio (r-O2:C) is compared across different contexts, the manuscript lacks further interpretation of what these deviations represent. The authors should also consider and discuss alternative explanations, including possible sources of error or variability in the observed ratios.

AUTHOR REPSONSE

>>> Thank you and we appreciate this concern. We have made a strong effort to improve the whole manuscript in how we discuss the implications of a variable ratio and what that could mean for oxygen cycling in coastal environments. Here we are highlighting three main ways we accomplish this (note in our below responses we include text from the current draft that could also be applied to this concern):

- We have strengthened our arguments that increased $r_{-O2:C}$ is representative of elevated oxygen consumption in deeper waters – given the same carbon flux. $r_{-O2:C}$ in coastal particulate organic matter is found to be 18% higher on average than organic matter formed in open ocean environments. Given a fixed carbon flux, it would require18% more oxygen to respire this material.
- A variable $r_{-O2:C}$ is also important for understanding the impact of climate change on coastal ecosystems. Many coastal ecosystems are vulnerable to anoxia. Our observations suggest that a variable $r_{-O2:C}$ adds to this vulnerability as future warming could result in a higher $r_{-O2:C}$ and hence a higher demand for oxygen in subsurface waters.
- We speculate that the variation in $r_{-O2:C}$ can come from a variety of sources. Our original manuscript finds a correspondence between environmental changes; ecosystem shifts and a variable $r_{-O2:C}$. We have now added additional discussion of how included various ecosystem processes including phytoplankton community composition, trophic processes of organic material and the presence to detrital material all can play an important role (see major concern #2 responses for direct details).

2. Community Structure vs. POC and $O_2$:C Ratios

The relationship between changes in phytoplankton community composition and the corresponding shifts in oxygen and carbon ratios (e.g., $O_2$:C) should be more thoroughly examined. An alternative perspective is worth considering: if community structure changes, would the composition and characteristics of particulate organic carbon (POC) also change? When applying a fixed POC conversion factor, how might shifts in species composition affect the resulting POC concentrations?

AUTHOR REPSONSE

>>> We agree with the reviewer that community composition will impact the characteristics of POC and subsequently $r_{-O2:C}$. Shifts in the POC quantity and quality stem from shifts in community composition and ecosystem processing. Previously, we have hypothesized an allometric relationship with $r_{-O2:C}$ such that cells with a larger surface-area-to-volume (e.g., cyanobacteria) has a higher $r_{-O2:C}$ compared to large cells with a small surface-area-to-volume (e.g., diatoms and dinoflagellates) (Moreno et al. 2022). In our current manuscript, we adjusted the discussion paragraph on community composition to include this hypothesis and more directly address how changes in ecosystem composition and functioning affect $r_{-O2:C}$. Our discussion now reads as follows:

"Our data suggests that $r_{-O2:C}$ is directly influenced by the community structure (Fig.4). Large phytoplankton have been hypothesized to have a lower $r_{-O2:C}$ on average compared to small phytoplankton (Moreno et al. 2022). We estimated the average functional group $r_{-O2:C}$ based on an allometric relationship with molecular components- primarily lipid, carbohydrate, and protein concentrations as cell grow (Moreno et al. 2022). Here, we found that $r_{-O2:C}$ was high in years where the relative abundance of 'Other Diatoms' and 'Other Dinoflagellates' are high (2016 and 2020). A missing factor in estimating $r_{-O2:C}$ is a cell's ability to store material. For example, diatoms can store a variety of compounds as needed under stress, including a variety of lipid compounds (Jallet et al. 2020). An increase in lipid storage will increase their $r_{-O2:C}$ range and add variability. Although dinoflagellates are estimated to have an average $r_{-O2:C}$ (average size and protein rich flagella), they also have storage capabilities. Similar to diatoms, during

stress an increase in lipid storage in dinoflagellates will increase their $r_{-O2:C}$ range. Shifts in physiological response by larger phytoplankton under stress could impact variable $r_{-O2:C}$ in this region. Additionally, changes in cyanobacterial ecotypes could also affect $r_{-O2:C}$. Previous time-series studies of cyanobacteria have demonstrated interannual patterns in ecotype relative abundance and seasonal switching in ecotype occurring in response to rapid environmental changes (Tai and Palenik 2009; Malmstrom et al. 2010; Nagarkar et al. 2018). Although ecotypes are genetically similar, physiologically and morphologically differences exist. Additionally, intracellular metabolites that are involved in methylating DNA, RNA, and proteins differ in culture and effect metabolism differently per ecotype (Kujawinski et al 2023). Under nutrient replete conditions *Prochlorococcus* and *Synechococcus* C:P and N:P ratios range from 121 – 165 and 21 -33 (Bertilsson et al. 2003), demonstrating significant variation in cellular composition. Further, cellular quotas of *Prochlorococcus* and *Synechococcus* in the North Atlantic varied significantly demonstrating species diversity and size distributions within the surface community (Baer et al. 2017). Although we do not quantify cyanobacteria in this study, previous published work at the MICRO site with a short overlapping window agrees that an ecotype shift occurred (Larkin et al. 2020). Our 2016 to 2018 $r_{-O2:C}$ ratios overlap with the Larkin data (Fig. 4) and as expected $r_{-O2:C}$ is highest in 2016 when *Prochlorococcus* HLI is in highest abundance. *Prochlorococcus* HLI dominates the cyanobacterial community under high temperature and lower nitrate concentration, whereas *Synechococcus* IV dominate in lower temperatures and higher nitrate (Larkin et al. 2020). As *Synechococcus* IV dominates the region, the $r_{-O2:C}$ ratio decreases. The shift in dominating cyanobacteria under its optimal conditions could result in a shift in the bulk chemical composition, in turn the POC concentration and $r_{-O2:C}$. We suggest that a shift in environmental conditions will impact the surface community composition and could add variation in the $r_{-O2:C}$."

There is a need to clarify the direction of causality: is the chemical composition driving community shifts, or vice versa? The current version appears to infer changes in community structure from observed chemical outcomes. However, it is equally plausible that structural shifts in the plankton community alter the stoichiometric ratios. This relationship should be explicitly discussed.

AUTHOR REPSONSE

>>> Overall, we completely agree with this sentiment. In the case of $r_{-O2:C}$ in this system, we think of it as one-directional. Shifts in community composition and ecosystem processing lead to shifts in organic C and in turn sets $r_{-O2:C}$. The community composition can be set or shift based on environmental factors. If the community shifts from a relatively small portion of diatoms to a higher proportion, then the $r_{-O2:C}$ will decrease based on our allometric relationship hypothesis leading to less oxygen consumption. In the above comment, we have included text which addresses this comment as well.

Furthermore, beyond community composition, the authors should also address how food web structure may influence the $O_2:C$ ratio. Is this ecosystem predominantly bottom-up or top-down controlled? Could shifts in trophic structure—not just primary producers—impact $O_2:C$ ratios? For example, changes in grazing pressure or predator-prey dynamics may also play a role and should be acknowledged.

AUTHOR REPSONSE

>>> Our study site is a highly dynamic coastal ecosystem that can experience top-down and bottom-up controls depending on season. There are two impacts that trophic structure can have. The first is the presence of zooplankton and bacteria, which makes up ~50% of total biomass and phytoplankton make up the other half with seasonal variation (Martiny et al. 2016). As the tropic structure shifts, this will be seen in the organic C present and will have an impact on respiration quotient. Respiration quotient is a measure of POC and its bonds, oxidation state, etc. As such, any impact on the quality and quantity of surface POC

will impact $r_{-O2:C}$ variability. We have included the following paragraph to the discussion to highlight this feature:

"Trophic structure and detrital matter could also influence $r_{-O2:C}$ variability. Zooplankton and heterotopic bacteria can makeup ~50% of surface organic matter (Martiny et al. 2016). Additionally, temporal POC variability exists in this region where 90% of POC was associated with unknown fractions- i.e., not planktonic- during certain periods (Martiny et al 2016). The relative presence and abundance of heterotrophic organisms are important in breaking down particles and setting POC concentrations. For example, POC is comprised of ~ 5 to 30% lipids (Wakeham et al. 1997; Hedges et al. 2001; Behrendt et al. 2024) and up to 25% detrital matter (Kaiser & Benner 2008, Martiny et al. 2016). In a recent study, bacterial degradation rates differed based on lipid content and bacterial preference (Behrendt et al. 2024). Some bacteria were highly selective and targeted specific lipid classes, whereas others did not (Behrendt et al. 2024). Preferential remineralization can occur (e.g., particulate organic phosphorus (Gundersen et al. 2002) and phosphonates (Benitez-Nelson et al., 2004). While the physiological mechanisms for this preferential remineralization are still being research, its presence could have strong impacts on the fate of detrital matter in surface waters, particularly in dynamic coastlines. These variations in POM remineralization and thus, oxygen consumption, can be reflected in the $r_{-O2:C}$. Similarly, the quality and quantity of detrital matter captured in the POM will also have an impact. Although, regional shifts in the fraction and quality of the detrital matter are currently unknown, previous estimates found that detritus and terrestrially derived particles make-up a small contribution to POC (Martiny et al. 2016). However, as grazers messily breakdown organic matter, produce fecal pellets, and reproduce, the composition of POC is changing. A shift in organic C caused by trophic interactions and/or detrital matter will result in a one directional shift in $r_{-O2:C}$."

3.  Implications and Biogeochemical Meaning in the Conclusion

The concluding section could benefit from a stronger emphasis on the broader biogeochemical implications of the observed elemental ratios. In particular, the manuscript should more clearly articulate how these findings inform our understanding of ecosystem function or carbon cycling under changing environmental conditions. Additionally, potential extensions of this work or new hypotheses that emerge from these patterns would enrich the conclusion.

AUTHOR REPSONSE

>>> We agree the conclusion needed a stronger emphasis of the broader implications. We have adjusted the conclusion to include how changing respiration quotient will cause a shift in oxygen. We also include the necessity for further research to be accomplished in coastal regions where decreasing oxygen can be determinantal. In addition to further research in coastal regions, we also discuss the need for mechanistic understanding of nutrients on $r_{-O2:C}$, as well as quantifying $r_{-O2:C}$ during HAB events. The conclusion reads as follows:

"Our study suggests that the $r_{-O2:C}$ is controlled by plankton's response to nutrient stressors in this coastal ecosystem. High $r_{-O2:C}$ along the coast has implications on future coastal hypoxic region expansions and general ecosystem health. Here, increased $r_{-O2:C}$ could lead to increase oxygen consumption by grazers and bacteria. Given future rising surface ocean temperatures (Di Lorenzo et al. 2005; Durack et al. 2018; Kwiatkowski et al. 2020; Xu et al. 2022) and continued changes to nutrients, $r_{-O2:C}$ dynamics further the complexities of coastal deoxygenation. Both direct (nutrient concentrations) and indirect (community shifts) controls may play important roles in setting $r_{-O2:C}$ and therefore future oxygen levels. Although much research is still needed to understand the mechanistic responses to nutrients and its effects on coastal oxygen levels, changes to community structure appear to have an impact on variable $r_{-O2:C}$ in this dynamic region. As shown in our study the average $r_{-O2:C}$ is 18% higher than open ocean

measurements. With increased temperatures and decreased nutrients due to climate change, we may observe increased coastal deoxygenation events at a higher frequency than in the open ocean. The potentially devastating impacts of increased events threaten the health and livelihood of surrounding communities that rely on tourism, fishing, and other ecosystem services. The respiration quotient is a grazer proxy for oxygen consumption during the breakdown of organic C. Ultimately, the MICRO $r_{-O2:C}$ is higher than previously measured open ocean samples. Our findings have strong implications for future respiration and oxygen cycling. Future research is needed to examine other coastal environments under a changing climate to understand the severity of coastal deoxygenation and determine if this study site is anomalous."

Minor comments:

Sampling periods are unclear in the Methods. By any vessels or boats. Are there any differences between each period as mentioned Line 172.

AUTHOR REPSONSE

>>> All samples are collected weekly from the Pier using a bucket. We added, "… off the Newport Pier using a bucket." to Line 91.

Line 95, "300 ml for POC and PCOD" is unclear. Each for 300 ml or a total of 300 ml?

AUTHOR REPSONSE

>>> We filter 300 ml of seawater for each parameter- so 300 ml for POC and 300 ml for PCOD. This is done in triplicates. We have adjusted the text to make this clearer. "For particulate organic carbon (POC) and for particulate chemical oxygen demand (PCOD) samples, we collect 300 ml for each analysis in triplicate from each bottle are filtered within an hour of collection through pre-combusted (500˚C, 5 hr.) 25 mm GF/F filters (Whatman, MA)."

**Reviewer # 2**

Moreno et al. present an analysis of respiration quotient (RQ) dynamics in the southern California Bight. The study reveals substantially higher RQ values in this coastal region compared to those typically observed in the open ocean. The authors also identify a strong inverse relationship between RQ and nitrate concentrations, highlighting the underlying biogeochemical drivers of RQ variability. Additionally, they discuss the potential influence of shifts in community composition and episodic oil spills on RQ patterns.

Overall, the manuscript is well written, logically organized, and the conclusions are convincingly supported by the long-term field observations. I have only a few minor comments for the authors' consideration:

AUTHOR REPSONSE

>>> We appreciate these supportive comments.

Line 57: I recommend adding a map of the study area (e.g., in the Supplementary Material) to improve clarity.

AUTHOR REPSONSE

>>> Thank you. We agree adding a map of our sampling site is beneficial. This is within the Supplementary Material (Figure S1)- we have adjusted all other figure labels to reflect this change.

Line 148: Table S1 appears to be missing.

AUTHOR REPSONSE

>>> Apologies. This has been fixed.

Line 184: Consider moving the two lower panels of Figure 1 downward and enlarging the markers and axis labels, as the current figure is difficult to read.

AUTHOR REPSONSE

>>> Thank you, we have adjusted the Figure 1 to be a 2 x 2 set up.

Line 187: Please clarify how "clear" is defined—can it be quantified?

AUTHOR REPSONSE

>>> We have applied statistical analysis (1 way ANOVA) to add a p-value value. We added " … annual differences (ANOVA p-value <0.05; Fig. S2)."

Line 188: Indicate in the caption that the squares in Figure S2 denote statistical significance.

AUTHOR REPSONSE

>>> We have added in the Figure S3 (previously S2) description "White squares represent statistical significance at p-value < 0.05."

Line 212: Justify the use of a 12-point moving average—does this correspond to a 2-month smoothing interval, and why was this window chosen?

AUTHOR REPSONSE

>>> Thank you for this comment. After looking back at our code, this was an error. Previously the 12-point moving average constrained the variability in the $r_{-O2:C}$ ratio the best, which represents a seasonal smoothly. However, we ultimately went with the rlowess moving average because seasonality in $r_{-O2:C}$ is present but not a strong statistical significance in our dataset. Additionally, our POC and PCOD red lines represent an 8-point moving averaging similar to the environmental data. We have adjusted our Figure 2 description as follows, ". The red line represents an 8-point moving average for A, B, and MATLAB 'rlowess' for C moving average. The respiration quotient is a molar ratio."